



**Simulating soil atmosphere exchanges and CO₂ fluxes for a restored**
**peatland**
Hongxing He[1]; Ian B. Strachan[2]; and Nigel T Roulet[1]
hongxing.he@mcgill.ca; ian.strachan@queensu.ca; nigel.roulet@mcgill.ca
Hongxing He, https://orcid.org/0000-0003-4953-7450
Ian Strachan, https://orcid.org/0000-0001-6457-5530
Nigel T Roulet, https://orcid.org/0000-0001-9571-1929
[1] Department of Geography, McGill University, Montréal, Quebec, Canada
[2] Department of Geography and Planning, Queen's University, Kingston, Ontario, Canada
Correspondence, HH hongxing.he@mcgill.ca; hongxing-he@hotmail.com
**Abstract**
Restoration of drained and extracted peatlands can potentially return them to carbon (C) sinks,
thus acting as significant climate change mitigation. However, whether the restored sites will
remain C sinks or switch to sources with a changing climate is unknown. Therefore, we adapted
the CoupModel to simulate soil atmosphere exchanges and the associated ecosystem CO₂
fluxes of a restored bog. The study site was a peatland in eastern Canada that was extracted for
eight years before restoration. The model outputs were first evaluated against three years
(representing 14-16 years post restoration) of eddy covariance measurements of net ecosystem
exchange (NEE), surface energy fluxes, soil temperature profiles, and water table depth data.
A sensitivity analysis was conducted to evaluate the response of the simulated CO₂ fluxes to
the thickness of the newly grown mosses. The validated model was then used to assess the
sensitivity of changes in climate forcing. CoupModel reproduced the measured surface energy
fluxes and showed high agreement with the observed soil temperature, water table depth, and



NEE data. The simulated NEE varied slightly when changing the thickness of newly grown
mosses and acrotelm from 0.2 to 0.4 m but showed significantly less uptake for a 1 m thickness.
The simulated NEE was $-95 \pm 19$ g C m$^{-2}$ yr$^{-1}$ over the three evaluation years, and $-101 \pm 64$ g
C m$^{-2}$ yr$^{-1}$, ranging from -219 to +54 g C m$^{-2}$ yr$^{-1}$ with an extended 28-year climate data. After
14 years of restoration, the peatland has a mean C uptake rate similar to pristine sites, but with
a much larger interannual variability, and under dry years, the restored peatland can switch
back to a temporary C source. The model predicts a moderate reduction of $CO_2$ uptake, but still
a reasonable sink under future climate change conditions if the peatland is ecologically and
hydrologically restored. The ability of CoupModel to simulate the $CO_2$ dynamics and its
thermal-hydro drivers for restored peatlands has important implications for emission
accounting and climate-smart management of drained peatlands.

**Keywords**: Restored peatland; climate variability; net ecosystem exchange; water table
depth; emission factor; simulation



## 1 Introduction

Degradation of peatlands through land use change and drainage is currently estimated to emit ~ 4% of global annual anthropogenic carbon dioxide (United Nations Environment Programme, 2022). Therefore, restoring drained peatlands so that they return to carbon sinks has been identified as an emerging priority for climate change mitigation (Leifeld and Menichetti, 2018). When ecologically restored successfully, peatlands can generally return to their carbon (C) uptake function after a decade or two following the recolonization of peatland vegetation and a decrease in water table depth (Nugent et al., 2018; González and Rochefort, 2014; Richardson et al., 2023; Tuittila et al., 1999; Wilson et al., 2016; Beyer and Höper, 2015). However, the C uptake function of restored peatlands is sensitive to climate conditions, particularly in drier years (Wilson et al., 2016). Therefore, changing climate can potentially weaken the sink strength or even switch the restored peatlands into C sources.

In North America, about a quarter of drained peatlands that were earlier used for horticultural peat extraction have been restored by the Moss Layer Transfer Technique (MLTT) (Chimner et al., 2017; Quinty and Rochefort, 2003). Ecosystem-scale flux measurements indicate peatlands remain a $CO_2$ source (~200 to 500 g C $m^{-2}$ $yr^{-1}$) the first few years of restoration (Petrone et al., 2001; Petrone et al., 2003), but after a decade or two, peat vegetation recovers, and the restored bogs return to C sinks with uptake rates similar to pristine sites (Nugent et al., 2018). While the C accumulation function can be fully restored within a decade or two, full restoration of the peat soil structure and ecohydrology takes a much longer time (Loisel and Gallego-Sala, 2022) with centuries to millennia required for the restored peatland to accumulate the C that was extracted. Restoration creates a novel ecosystem in transition to a rewetted steady state and the altered ecohydrology decreases peatland ecological resilience (Kreyling et al., 2021).




The ecological function of peatlands is strongly linked to ecohydrology (Waddington et al.,
2014). Recently, He and Roulet, (in review) applied the conceptual four functional layers of
peatlands (i.e., green; peat litter; collapse; peat proper) introduced by Clymo (1992) to two
well-studied bog sites in eastern Canada, and showed a wider range of water table fluctuation
and a larger frequency of water table drops below the annual mean in the restored bog compared
to that of the pristine bog, mainly due to the lack of the mesotelm collapse layer (Clymo and
Bryant, 2008). The newly regenerated moss with low bulk density forms large pores directly
above the dense residual peat remaining after extraction (catotelmic peat) and does not have
the negative interstitial pressures required to draw pore water, causing a capillary barrier effect
(Gauthier et al., 2022; Gauthier et al., 2018). The capillary barrier decreases the ability of the
new moss to draw water from the deeper compacted catotelmic peat, thus causing an overall
lower surface moisture content for restored sites compared to natural peatlands (McCarter and
Price, 2015). As a result, the new moss layer may become stressed quickly during dry periods.
Synthesis studies have shown that vegetation colonization is much slower after restoration over
warm and drier years (González and Rochefort, 2014), and data from a restored Irish extracted
bog show a less resilient C uptake function over the drier years (Wilson et al., 2016).

Under the United Nations Framework Convention on Climate Change (UNFCCC), countries
with peatlands managed for extraction are required to report greenhouse gas emissions annually
(IPCC, 2014). However, $CO_2$ uptake and/or emissions from the restored peatlands so far have
not been accounted for in Canada (National Inventory Report of Canada; (ECCC, 2021) mainly
due to the large uncertainty in the emission factor (EF) calculation. Currently, there is a
discussion that restoration can create C credits and thus could be used to offset the C emissions
during the drainage phase (Tanneberger and Wichtman, 2011). The IPCC report for restored





peatlands uses default EFs (i.e., Tier 1) based on literature data (IPCC, 2019). An emission
factor based on empirical observations (i.e., Tier 2) offers improvement as it is subject to the
environmental conditions and the time of year the measurements were done. Yet, most of the
observed data is of short duration and thus can not capture interannual variations in emissions
and associated environmental variables. Process-based modeling of restored peatlands (i.e.,
Tier 3) can be used to determine the 'representativeness' of the empirical EFs by examining
the coupled hydrological-C dynamics and how they vary over within and between years. He
and Roulet (2023) showed that directly using literature data to generate emission factors can
be biased because it does not account for seasonality and interannual climate variability.

Existing studies using models for restored peatlands are few. Lees et al. (2019) applied a
satellite-based, temperature-driven gross primary productivity (GPP) model over peatland sites
at various stages of restoration in the UK and Ireland and found that the model can simulate
the GPP measured by eddy covariance. Premrov et al. (2021) modified the drainage function
in the ECOSSE model to simulate the water table and $CO_2$ flux for drained and rewetted
extracted bogs, but their model evaluations showed that ECOSSE still requires further
development to accurately simulate the water table depth for the rewetted sites. Recently,
Lippmann et al. (2023) introduced a dynamic vegetation scheme in the PVN model, driven by
input water table data, and evaluated the model for the measured $CO_2$ flux together with the
vegetation competitions in two restored nutrient-rich peatlands in the Netherlands. However,
none of these models consider the coupled ecohydrology and C dynamics for restored peatlands
(Silva et al., 2024). Previous research showed that CoupModel could successfully simulate
peatland $CO_2$ dynamics associated with various land-use options (e.g., drained peatlands for
forestry; (He et al., 2016a; He et al., 2016b; Kasimir et al., 2021), land-use change of afforested
peatlands (Kasimir et al., 2018) and five European peatlands with various land-uses, including



restored sites (Metzger et al., 2015). Recently, the model was applied to simulate the $CO_2$ fluxes
of a pristine continental bog (He et al., 2023a) and an active peat extraction site (He et al.,
2023b). These studies provide a basis for further use of the model to simulate restored peatlands
to close the land use cycle from pristine peatlands, drainage for different land uses to final
restoration.

The overall aim of this study is to simulate the soil-atmosphere exchanges of heat, water, and
$CO_2$ fluxes for a bog restored by the MLTT technique. More specifically, we aim to:
1) adapt and evaluate the CoupModel to simulate net ecosystem exchange (NEE) and its hydro-
thermal drivers, including surface energy fluxes, soil temperature profile, and water table depth;
2) test the model sensitivity to varying thickness of newly grown mosses and the acrotelm;
3) evaluate the impact of interannual climate variability on the simulated ecosystem $CO_2$ flux
and discuss its implications for emission factor calculation; and,
4) predict the impact of future climate change on the C uptake function of restored peatlands.

**2 Site and methods**
**2.1 Site Description**
The Bois-des-Bel (BDB) peatland is located 11 km northeast of Riviere-du-Loup, Quebec
(47°58'1.95"N 69°25'43.10"W). The peatland complex covers an area of 202 ha. A small sector
of 11 ha was extracted for horticulture peat by vacuum harvesting between 1972 and 1980. In
the autumn of 1999, an 8.1 ha area was restored using the MLTT. The climate of the region is
cool-temperate with an average long-term (1981-2010 climate normal St-Arsene) annual
temperature of 3.5 ºC and annual precipitation of 962 mm (Environment and Climate Change
Canada, 2023). BDB is well studied and detailed descriptions of the restoration procedure and
site characteristics can be found in several publications (McCarter and Price, 2015; Strack and





Zuback, 2013; Waddington and Day, 2007; Poulin et al., 2013). Nugent et al. (2018) measured
the soil-atmosphere exchanges by eddy covariance between 2013-2016, i.e., 14-17 years after
the restoration. In this study, we used their measured meteorological data (Table 1) for model
forcing, and measured water table depth, peat temperatures, and flux data for model evaluation.

**2.2 Brief Model Description**


The CoupModel (coupled heat and mass transfer model for soil–plant–atmosphere systems)
platform is a process-based model designed to simulate water and heat fluxes, along with the
C-N-P cycle, in terrestrial ecosystems (Jansson, 2012; He et al., 2021). The main model
structure is a one-dimensional multi-layered soil profile. Model forcing is measured weather
data (Table 1). The model and technical description are freely available at
www.coupmodel.com. CoupModel was previously applied to simulate ecohydrology and $CO_2$
exchanges for a pristine bog, Mer Bleue that resembled, though with fewer trees, the BDB site
before opening for extraction (He et al., 2023a), and recently successfully simulated one
ongoing peat extraction site, Riviere-du-Loup in the same region as BDB (He et al., 2023b).
The setup and model structure of the BDB simulation were thus built on the base of the upper
aerobic peat layer and vegetation characteristics of Mer Bleue and the residual extracted peat
layer of Riviere-du-Loup. Here, we report the model setup unique for the BDB site. More
detailed process descriptions, model structure, and parameters are reported in He et al. (2023b)
and He et al. (2023a).

**2.3 Simulation Design, Model Structure, Initial and Boundary Conditions**


CoupModel was used to simulate the soil vegetation processes and linked hydrology and
energy flows of BDB in a 30-minute time-step from 2013-07-14 to 2016-11-01.



Nugent et al. (2018) conducted a detailed vegetation survey at BDB in 2013 and these data
were used to initialize the vegetation conditions in CoupModel. The survey showed *Sphagnum*
mosses and *Polytricum strictum* cover more than 90% of the surface with a new acrotelm
thickness of ~ 0.3 m, sedges (*Eriophorum vaginatum* and *Carex* spp.) cover 33%, and
ericaceous shrubs (*Chamaedaphne calyculata*, *Rhododendron groenlandicum*, *Kalmia*
*angustifolium*, *Vaccinium oxycoccus*, and *V. angustifolium*) cover 39% of the soil surface
(Nugent et al., 2018). Trees (*Picea mariana* and *Larix laricina*) were few but were also
beginning to expand across the site. *Typha latifolia* from the remnant ditches covers 4% of the
total site area. In our simulation, we grouped vegetation into three plant functional types or
modeled vegetation layers: the first group represents the Ericaceous shrubs and the trees, that
cover ~40% of the surface, with an assumed lowest root depth of 0.5 m. The second group
represents the sedges, which cover 33% of the surface and lowest root depth of 0.35 m. The
third group represents the *Sphagnum* mosses and other non-vascular vegetation (*Ploytricum*
*Strictum*) at the soil surface which cover 90% of the soil surface with no roots. These three
modeled vegetation layers were described in the model using the "multiple-big-leaves" concept
considering dynamic competition in terms of interception of light and uptake of water. For each
vegetation layer, plants were conceptually divided into leaf, stem, coarse root and fine root.
For the moss layer, the live capitulum was conceptually viewed as leaf and the rest as stem in
the model (He et al., 2023a). C and the dynamics of the plant development are simulated as the
interactions between plant and physical driving forces; e.g., how the plant cover influences
both aerodynamic conditions in the atmosphere and the radiation balance at the soil surface.
Since these are oligotrophic ecosystems, the influence of nutrients on C was not considered in
this study. The three vegetation groups were pre-run for fourteen years to spin up and reach a
quasi-steady state (defined as no abrupt takeover or die-offs of one vegetation group).





For the peat soil, we simulated the first 1.8 m of peat in BDB which includes 0.3 m of the
surface newly developed acrotelm and mosses and 1.5 m of the residual extracted peat. We
divided the peat soil profile into nine layers: from 0.05 m per layer at the top to 0.80 m per
layer at the bottom. For each simulated layer, the peat soil water retention curve and unsaturated
hydraulic conductivity were estimated by the Mualem-van Genuchten model (Mualem, 1976;
van Genuchten, 1980). The physical and hydraulic properties used in this study were compiled
from the measured data from BDB (Table 2). Water flow between soil layers follows Darcy's
law as generalized for unsaturated flow by Richards (1931). We additionally simulated bypass
flow to account for preferential water flow in the root channels, and macro-pores by using an
empirical bypass flow scheme (Jansson et al., 2004). Soil heat flow between soil layers was
assumed to be mainly driven by conduction. CoupModel solves water and heat equations
simultaneously within the soil-plant-atmosphere continuum, and water and heat are coupled in
a dynamic way to the plant vegetation layers; accounting for feedback interactions between the
plant and the environment.

The initial conditions for water and heat were from measured data (Nugent et al., 2018). The
initial condition for soil C stocks for each soil layer was calculated from the measured bulk
density and C concentration (assumed 50%). The total C in the 1.8 m soil profile was 101.8 kg
C m$^{-2}$ (Table 2). Similar to He et al. (2023b), we used two soil C pools which differed in
substrate quality and hence decomposition rate to model the impact of organic matter quality
on soil respiration: labile and refractory soil C. The partitioning ratio between these two pools
from Riviere-du-Loup was used for the bottom 1.5 m at BDB, while for the top 0.3 m of newly
grown peat, 80% was assumed to be in the labile pool. The decomposition rate coefficient
(Table S1) and its response to temperature and water were kept the same as He et al. (2023a).





We assumed no vertical water flow for the lower boundary condition (i.e., at 1.8 m depth) due
to the very low saturated hydraulic conductivity (Table 2) and assumed a small thermal heat
flow across the lower boundary condition for heat. The site was also drained laterally to the
outflow at a distance of ~200 m (Shantz and Price, 2006). The model parameter values were
primarily obtained from the measured data, and where not available, literature values used in
previous model applications were applied (Table S1).

**3 Results**
**3.1 CoupModel evaluation for restored peatland**
CoupModel simulated the half-hourly surface energy balances well, as shown by the high
agreement with the measured total radiation, sensible and latent fluxes (coefficient of
determination, $r^2$ >0.7 for all, Figs. 1a, b, c), and surface soil heat flux ($r^2$ =0.4, Fig. 1d).
However, the model tended to overestimate the sensible heat flux and underestimate the latent
heat flux, particularly over the periods of spring and earlier summer, where the model simulated
a smaller and delayed (~ 1 month) increase of latent heat fluxes compared to the measured data
(Fig. S1). The lower agreement with soil surface heat flux is due to its residual energy flux,
thus small in flux size, i.e., one order of magnitude lower compared to the turbulent energy
fluxes (Figs. 1d and S1), plus the energy balance closure calculated with measured data over
the three years is ~90% while CoupModel has full energy conservation.

The model simulates the measured soil temperature profile over 5-20-80 cm depth generally
well, with $r^2$ >0.9 for all three soil layers, (Figs. 1e, f, g) however, the model showed difficulty
in precisely simulating the soil thawing (i.e., zero curtain effect Fig. S2). The simulated
temperature started to increase above zero a half month earlier than did the measured data for
the 20-80 cm depth in 2015 but was delayed for almost one month for 2016 (Fig. S1).





CoupModel probably overestimated the soil frozen depth as higher heat flow was partitioned
into the soil surface over May to June every year (Figs. 1d, S1), thus extra heat was needed for
thawing in the spring and delayed the increase of latent heat fluxes and temperature increase.

Model performance for water table depth was generally less good compared to the energy and
temperature variables. However, the model still captured 50% of the measured variations ($r^2$
=0.5, Fig. 1h). CoupModel generally simulated a smaller magnitude fluctuation compared to
the measured data and the model data agreement was better over the summer than the winter
(Fig. 2a). For instance, large infiltration from snow melt around May was simulated in the
model every year, but not represented in the measured data, probably again reflecting the
model's difficulty in precisely capturing the phase change over winter (Fig. 2a).

Measured daily net ecosystem exchange data ranges from ~ -3 g C m$^{-2}$ d$^{-1}$ (negative indicating
uptake) during July to a loss of ~ +2 g C m$^{-2}$ d$^{-1}$ during cloudy days or shoulder seasons (Fig.
2b; Note that the flux data is 30 min in Figures 1i and 2b). CoupModel reproduced the measured
half-hourly NEE data reasonably well ($r^2$ = 0.64; Figs. 1i and 2b). Nugent et al. (2018) gap-
filled the BDB eddy covariance data and estimated an annual C flux of -90 ± 10 (± 95% CI), -
105 ± 7, and -70 ± 7 g C m$^{-2}$ yr$^{-1}$ in 2014, 2015 and 2016, respectively. The corresponding
simulated annual fluxes are -89, -120 and -75 g C m$^{-2}$ yr$^{-1}$, respectively. The model simulated
a delayed start of spring uptake during the years 2014 and 2016, which again can be explained
by the delayed thawing in the model.

**3.2 Sensitivity to the thickness of the newly grown mosses**
We conducted a sensitivity analysis to evaluate the model responses to the thickness of the
newly grown mosses (i.e., new acrotelm) which partly represents the time since the restoration.



It has been argued that -100 mb is the limiting soil moisture pressure head for sustaining moss
growth (McCarter and Price, 2012). Three extra model simulations were made based on the
reference run (30 cm acrotelm) with new acrotelm thicknesses of 20 cm, 40 cm, and 100 cm.
For the latter two model simulations, peat properties of the 20-30 cm layer in the reference run
(Table 2) were assumed for the future extra 10 and 70 cm acrotelm, respectively. The
vegetation was assumed to be the same as the reference run and the peat compaction due to the
growth of mosses and decomposition was not considered (i.e., no mesotelm collapse layer).
Our sensitivity analysis showed that the simulated NEE uptake increased slightly when
changing the new acrotelm thickness from 20 to 40 cm but reduced (meaning less uptake)
significantly for the model run with an acrotelm of 100 cm (Fig. 3). The small changes of
simulated NEE can be explained by both increase of GPP and ecosystem respiration (ER) with
increasing new acrotelm thickness (20-40 cm). The reduction of $CO_2$ uptake in the 100 cm
acrotelm thickness model run is because the model simulated that the surface mosses start to
die off because they can't take up water from the deep peat (Fig. 3).

**3.3 Interannual climate variability on $CO_2$ uptake of restored peatlands**
The BDB region shows large annual climate variability over the last 28 years from 1994 to
2021. The measured annual mean air temperature ranged from 2.6 to 5.7 $^{\circ}$C and the annual
precipitation from 633 to 1488 mm (Figs. 4a, b). This can be compared to the 30-year annual
mean air temperature of 3.5 ± 2.9 $^{\circ}$C and the precipitation of 962 mm for the climate normal
data (1981-2010) at St-Arsene station (Environment and Climate Change Canada, 2023). Both
annual air temperature and precipitation showed increasing trends over the measured period
from 1994 to 2021, with a slope of 0.03 $^{\circ}$C yr$^{-1}$ for air temperature and 1.69 mm yr$^{-1}$ indicating
possible future warmer and wetter conditions in the region. The weather over the three years

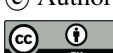



of flux measurement was generally around the mean climate conditions (for more discussion
see Nugent et al. (2018)).

We made an extra simulation (in daily time step) with a 28-year climate input based on the
2013-2016 BDB set up to represent the normal climate variability also including extreme years.
The simulated 28-year average of $CO_2$ uptake was -101 $\pm$ 64 g C m$^{-2}$ yr$^{-1}$, ranging from a
maximum uptake of -219 g C m$^{-2}$ yr$^{-1}$ in 1999 to a loss of +54 g C m$^{-2}$ yr$^{-1}$ in 2015 (Fig. 5). At
the annual scale, $CO_2$ uptake seems to increase slightly with increasing air temperature,
although the relationship was not statistically significant (p=0.19). The model simulated the
BDB peatland as an atmospheric $CO_2$ source for three years 1995, 1997, and 2015, all of which
had below-average precipitation.

We further compared the simulated flux rates with long-term measurements at Mer Bleue, a
pristine shrub-*Sphagnum* bog within the same climate region. Over fifteen years of
measurement (2004 to 2018), Mer Bleue had an average uptake rate of -108 $\pm$ 33 g C m$^{-2}$ yr$^{-1}$
(He et al., 2023a), similar to the three-year BDB uptake rate measured by the tower, -90 $\pm$ 18
g C m$^{-2}$ yr$^{-1}$, and the current 28 year extended simulation, -101 $\pm$ 64 g C m$^{-2}$ yr$^{-1}$ (Fig. 6).
Therefore, after fourteen years of restoration, the BDB peatland has switched back to C uptake
and the uptake rate was similar to pristine sites (for more discussion see Nugent et al. (2018)).
However, our model simulations additionally show that the C uptake at the restored peatland
has larger interannual variability (S.D. 64 g C m$^{-2}$ yr$^{-1}$) compared to the pristine Mer Bleue site.
Under certain dry years, the restored site can potentially switch back to C sources while the
pristine peatlands showed persistent C uptake with a smaller interannual variation (S.D. 33 g
C m$^{-2}$ yr$^{-1}$). In other words, the restored peatlands seem to have less ecological resilience
compared to the pristine peatlands.




### 3.4 Future climate change on $CO_2$ uptake of restored peatlands

We evaluate the potential of future climate change on the $CO_2$ uptake function of the restored
bog using the 28-year simulation as the long-term reference run. Climate change scenarios were
designed as combinations of an increase of all-year-round air temperature for the 28-year
climate data by +1, +2 °C, and/or change all-year-round precipitation by ±10%, the range of
climate change expected for this area of Quebec (Zhang et al., 2019). Then, equilibrium model
runs using the 2013-2016 BDB setup for the future climate, were conducted to evaluate the
potential response of C uptake functions.

Our model simulations show that increasing air temperature will decrease the $CO_2$ uptake rate
of restored peatlands. Increasing air temperature alone by 1 °C decreases the annual C uptake
rate by 5% compared to the reference run, and ~16% when air temperature increased by 2 °C
(Fig. S3). A larger rate of decrease under the +2 °C scenario can be explained by the simulated
more pronounced water table drop (Fig. S3). Our model simulation shows a change of ±10%
in precipitation alone only influences the $CO_2$ flux marginally, with a reduction of uptake rate
when precipitation decreases (Fig. S4). The BDB region is humid (annual
precipitation/potential evaporation ratio is ~ 1.5 to 2 (Hare and Thomas, 1979)). Thus, a 10%
change in precipitation is predicted to influence the water table marginally (Fig. S4). We made
a climate scenario with an increase of air temperature by 2 °C and reduced precipitation by
10%, i.e. the 'extreme' scenario. The restored bog still acts as a C sink overall, with a slightly
reduced (~ -6%) simulated mean uptake rate of -95 g C m$^{-2}$ yr$^{-1}$ (Fig. 7). The modified climate
causes both GPP and ER to increase (Fig. 7), thus effectively canceling each other out. Our
model simulations thus overall suggest the restored peatlands will likely maintain their C
uptake functions under future climate change.




**4 Discussion**

Current model evaluation with the dataset from the BDB site shows CoupModel can simulate
the coupled hydrology, heat, and $CO_2$ fluxes of a restored peatland. CoupModel has been
applied to Mer Bleue, a pristine bog (He et al., 2023a), and Riviere-du-Loup, an active peat
extraction site (He et al., 2023b). The ability of the model to simulate C dynamics associated
with ecohydrology for the restored system thus closes the land use cycle and shows the model
can now simulate all stages of land uses, from pristine peatlands, to drained for extraction and
finally restoration.

Our model performance for $CO_2$ flux is similar to previous models that have been applied to
restored sites, such as the ECOSSE model (Premrov et al., 2021) and the PVN model
(Lippmann et al., 2023). However, the advance of our current modeling exercise compared to
the earlier studies is its capability of accurately simulating both the water table depth and C
dynamics in a finer temporal resolution. CoupModel simulates the coupled C-hydrological
processes in half-hour resolution, while a daily time step was used for the earlier models. The
ability to simulate processes at a sub-daily scale is particularly important for the future
inclusion of $CH_4$ as the transport processes (e.g., ebullition) occur at a sub-daily scale (Walter
and Heimann, 2000). Empirical studies have shown that the water table is an important control
for greenhouse gas fluxes in restored peatlands (Evans et al., 2021; Järveoja et al., 2016; Koch
et al., 2023). Restoration is associated with management practices that change the hydrology
of the peatlands, such as blocking the drainage ditches at the beginning of restoration. With the
gradual recovery of peat vegetation and the development of the mesotelm collapse layer, the
water table fluctuations further reduce, and the mean level gradually moves above the
mesotelm (He and Roulet, in review). Therefore, following restoration, the ecohydrology and



vegetation co-evolve and feedback between each other, co-regulating the overall C uptake
function of the peatland. The ability of CoupModel to simulate the coupled processes thus has
important implications for understanding the overall climate impacts of peatland restorations.
Our study simulates the time frame of 14 to 16 years after restoration, representing a stage of
fully recovered vegetation. Future modeling research should cover the beginning of the
restoration thus simulating the full dynamic coupling of vegetation development, hydrology
management, and peat soil development.

The extended model simulations show that restored peatlands have less resilience to climate
variability than do pristine peatlands (Figs. 5 and 6). Theoretical studies have argued that bogs
are complex adaptive systems based on the tight feedbacks among plant production,
decomposition, and water storage represented by water table depth (Eppinga et al., 2009;
Belyea and Clymo, 2001). Due to the missing collapse layer, the ecohydrology of restored
peatlands is not fully restored. Water table frequency distribution can be a useful measure for
evaluating the success of ecohydrology restoration (He and Roulet, in review). CoupModel
predicts that the water table frequency distribution for BDB will gradually recover to a state of
a pristine bog when the newly grown mosses at the surface reach 40 cm depth (data not shown).
Our model sensitivity analysis shows that mosses cannot thrive under a 100 cm acrotelm
thickness which is in agreement with results from field studies that suggest a tension of -100
cm water as the hydrologic threshold for *Sphagnum* establishment (Price, 1998; Price and
Whitehead, 2001). The ability of CoupModel to reproduce such important ecohydrology
regulation has implications for future model applications to evaluate the impacts of field
management practices on greenhouse gas fluxes by changing boundary and lateral hydrology
conditions.



The current model exercise represents a series of studies towards developing CoupModel as an
IPCC Tier 3 methodology for estimating emissions from extracted and restored peatlands (He
et al., 2023b; He and Roulet, 2023). Our work to date has focused on bogs in eastern Canada
but should be expanded to include bogs and poor fens in western Canada and other
geographical and ecoclimate regions in the future. To date, there are few emission data from
restored peatlands, and those data are snapshots covering only a few years and thus do not
reflect the temporal dynamics of greenhouse gases (Kalhori et al., 2024). Our long-term model
simulations suggest an EF of $-1.01 \pm 0.64$ t C ha$^{-1}$ yr$^{-1}$ for a bog 14-16 years post restoration by
the MLTT; this is ~five times larger (meaning more uptake) than the default IPCC Tier 1 EF
for temperate nutrient-poor rewetted organic soils (-0.23 with CI -0.64 to +0.18 t C ha$^{-1}$ yr$^{-1}$
(IPCC, 2014). The data used to generate the IPCC EF includes more degraded sites in Europe
and different rewetting methods. The Canadian practice of leaving a residual peat layer at the
end of extraction and using MLTT for restoration seems to be beneficial for the recovery of
peatland C uptake. The default IPCC EF has earlier been used to evaluate the overall climate
impacts for peatland restoration using a modeling approach (Gunther et al., 2020). Our results
thus suggest those studies might significantly underestimate the climate cooling effects for
Canadian bog sites that have been restored using MLTT.

Our climate change simulations show the important regulating affect of air temperature on the
$CO_2$ uptake of restored peatlands, where future global warming is predicted to moderately
weaken the sink strength (Fig. S3). However, it should be noted that future changes in seasonal
patterns and extremes were not accounted for in our climate change scenarios. Helbig et al.
(2022) analyzed flux measurements from northern peatlands and showed earlier summer
warming increases NEE uptake while late summer warming decreases it. There is also the
possibility of fire that would structurally alter the peatlands. Our simulations do not include



fire, which is much less common in eastern Canadian peatlands than in the west (Zoltai et al.,
1998; Lavoie and Pellerin, 2007). Thus, our climate change simulations probably represent a
conservative prediction which might in turn explain the moderate reduction of sink strength.
As our extended simulations show, it is possible that over extreme years the site can switch to
a small $CO_2$ source and that potentially the number of source years could increase in the future.

**5 Conclusion**
This study applied the CoupModel to a peatland site restored 14-16 years previously. We
conclude:
•    CoupModel can describe the measured sub-daily $CO_2$ fluxes, hydrology, and heat for

424         the restored peatland system.

•    Restored peatlands have less resilience to climate variability than pristine peatlands.
•    CoupModel simulation results in an emission factor of -1.01 ± 0.64 t C ha$^{-1}$ yr$^{-1}$ for

427         Canadian bogs that have been restored for 14 to 16 years by the moss layer transfer

428         technique, ~ five times larger than the IPCC default emission factor.

•    CoupModel now simulates all stages of peat extraction and restoration, and can be used

430         for exploring land-use change issues, suggesting climate-smart management practices,

431         and Tier-3 emission reporting.




Table 1 Data from Bois-des-Bel peatland used for the CoupModel forcing and evaluation

| Category | Variable | Unite | Resolution | Period | n | References |
|---|---|---|---|---|---|---|
| Model forcing - meteorological data | Global solar radiation | J m⁻² d⁻¹ | 30 min | 2013-2016 | 59952 | Nugent et al. (2018) |
| | Air temperature | °C | | | | |
| | Relative humidity | % | | | | |
| | Precipitation | mm d⁻¹ | | | | |
| | Wind speed | m s⁻¹ | | | | |
| Evaluation data | Total net radiation | J m⁻² d⁻¹ | 30 min | 2013-2016 | 49964 | Nugent et al. (2018) |
| | Soil heat flux | J m⁻² d⁻¹ | 30 min | 2013-2016 | 56631 | |
| | Latent heat flux | J m⁻² d⁻¹ | 30 min | 2013-2016 | 23397 | |
| | Sensible heat flux | J m⁻² d⁻¹ | 30 min | 2013-2016 | 25511 | |
| | Soil temperature profile | | | | | |
| | 5-80 cm depth, thermocouples | °C | 30 min | 2013-2016 | 52892 | |
| | Water table depth | m | 30 min | 2013-2016 | | Nugent et al. (2018) |
| | Net ecosystem exchange | g C m⁻² d⁻¹ | 30 min | 2013-2016 | 18920 | |


Table 2 Physical, hydraulic, and Mualem-van Genuchten coefficients for Bois-des-Bel site

| Peat layer | Modeled layer (cm) | $\rho_B$ (g cm⁻³) | $\theta s$ (vol%) | $\theta r$ (vol%) | $\alpha$ | $n$ | $k_{sat}$ (mm d⁻¹) | C stock (g C m⁻²) |
|---|---|---|---|---|---|---|---|---|
| Newly grown mosses | 0-5 | 0.025 | 98.8 | 10 | 0.16 | 2.51 | 1×10⁵ | 625 |
| | 5-10 | 0.03 | 98.5 | 14 | 0.09 | 2.96 | 1×10⁵ | 683 |
| | 10-20 | 0.032 | 96 | 14 | 0.09 | 2.96 | 1×10⁵ | 1588 |
| | 20-30 | 0.04 | 95 | 10 | 0.09 | 2.96 | 1×10⁴ | 1888 |
| Residual extracted peat | 30 - 40 | 0.08 | 94 | 10 | 0.022 | 2.03 | 5×10³ | 4025 |
| | 40-50 | 0.13 | 91 | 20 | 0.016 | 4.05 | 4×10² | 6500 |
| | 50 - 70 | 0.1 | 93 | 20 | 0.025 | 1.39 | 2×10² | 9500 |
| | 70-100 | 0.13 | 90 | 30 | 0.013 | 1.4 | 2×10² | 21000 |
| | 100 - 180 | 0.14 | 90 | 30 | 0.008 | 1.45 | 6×10² | 56000 |


Bulk density $\rho_B$ , porosity $\theta s$ and saturated conductivity $k_{sat}$ data were from McCarter and Price (2013),
Gauthier et al. (2022) and Petrone (2002). Non-linear curving fitting was run with the empirical constant m=1-
1/n with the wilting point $\theta w$ set to 10 % for the topsoil layer, and 30% for the 40-150 cm layer (Menberu et al.,
440    2021).



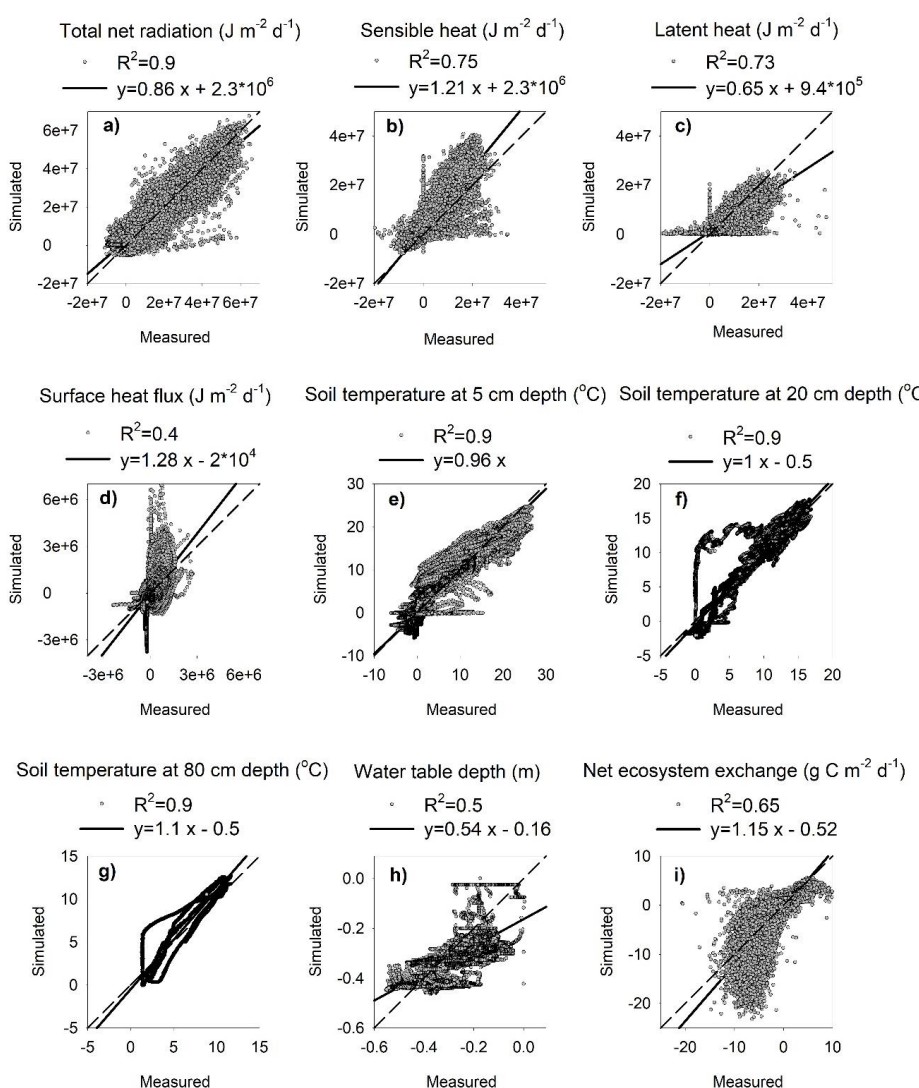

Figure 1 Relationship between simulated and measured 30-minute a) total net radiation, b) sensible heat, c) latent heat, d) soil surface heat flux, e) soil temperature at 5 cm depth, f) soil temperature at 20 cm depth, g) soil temperature at 80 cm depth, h) water table depth, and i) net ecosystem exchange over the period 2013 to 2016 (n=56600)



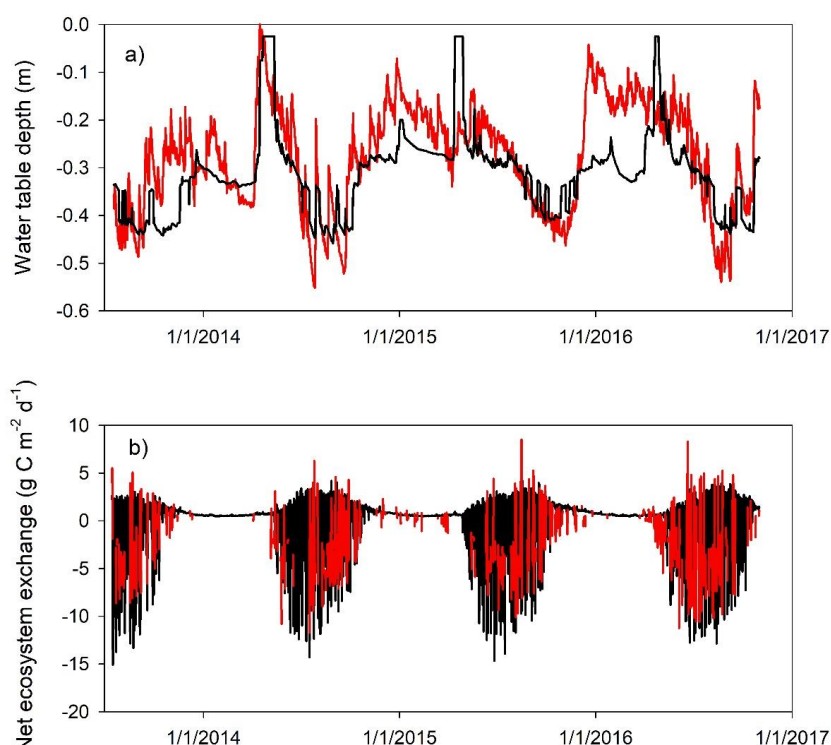


Figure 2 Measured (red) and simulated (black) 30-minute a) water table depth, b) net
ecosystem exchange.




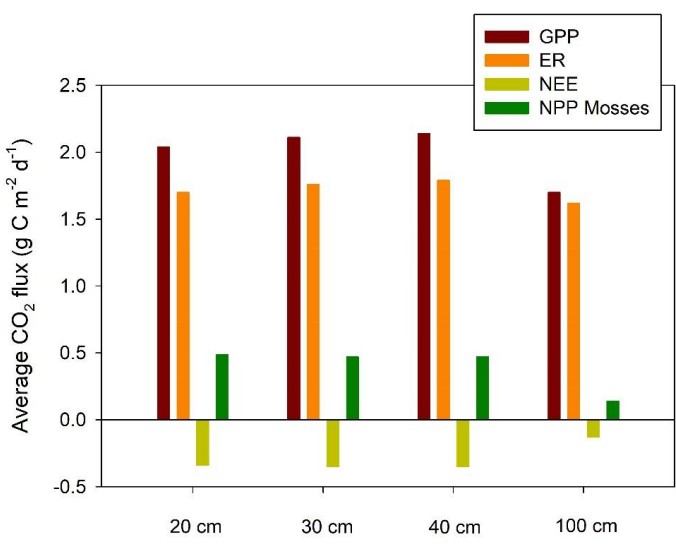


Figure 3 The response of simulated average ecosystem $CO_2$ fluxes (2013-2016) to the

simulated thickness of the newly grown mosses







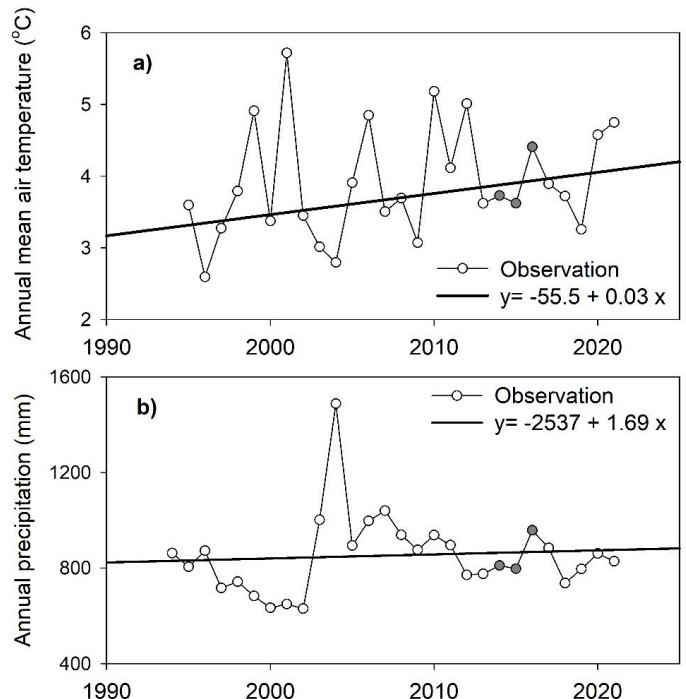


Figure 4 Variability in a) annual mean air temperature; b) annual precipitation between 1994
and 2021 as recorded at Rivière-du-Loup (ECCC historical climate data, 2022). The shaded
circles indicate the measured period of the eddy covariance tower.




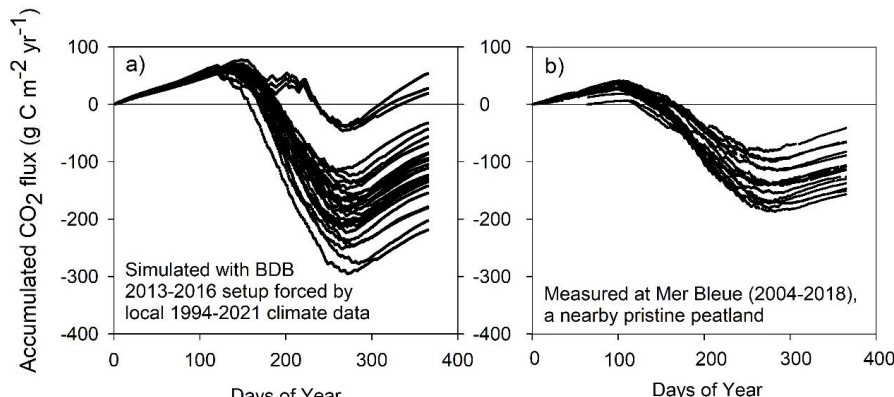


Figure 5 Accumulated annual $CO_2$ flux a) simulated with BDB 2013-2016 setup forced by

Rivière-du-Loup 1994-2021 climate data; b) measured over 2004-2018 at Mer Bleue (He et al.

2023b), a pristine peatland in the same climate region.









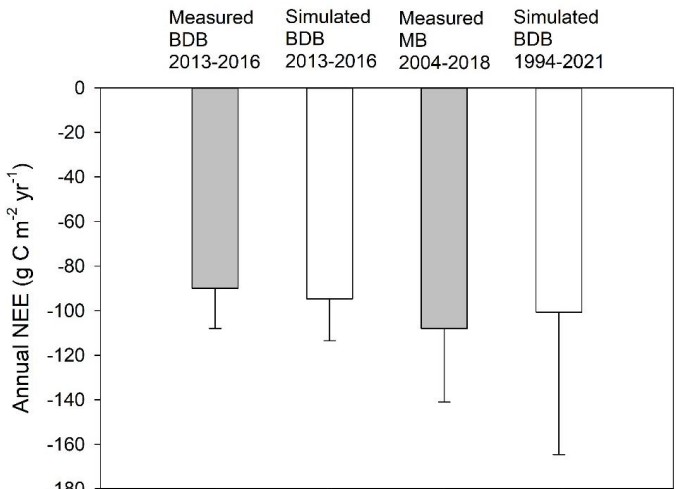


Figure 6 Comparison of $CO_2$ fluxes and emission factors from the different approaches.



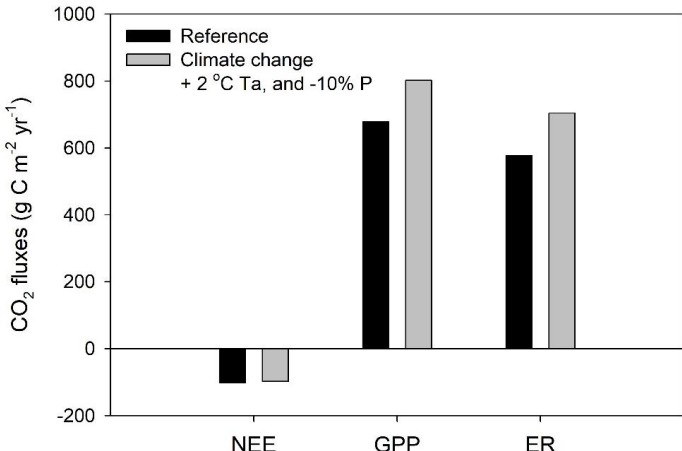


Figure 7 Simulated $CO_2$ fluxes under a scenario where air temperature is increased year

around by 2 °C and precipitation is decreased by 10%. Equilibrium model runs used the BDB

2013-2016 setup and Rivière-du-Loup 1994-2021 climate data.




Data Availability
The version of the CoupModel used to run the model simulations, including the source code
is hosted on Zenodo (https://zenodo.org/record/3547628) and the executed CoupModel is
available at www.coupmodel.com.

Author Contributions
HH and NR led the work, IBS led the eddy covariance data component, HH did the
modeling, analysis and drafted the paper with help from NR, all authors contributed to editing
and revision of the paper.

Competing Interests
The contact author has declared that none of the authors has any competing interests.

Acknowledgements
HH was supported by funding to NTR through the National Science and Engineering Research
Council of Canada's Collaborative Research and Development grant, in partnership with the
Canadian Sphagnum Peat Moss Association (CSPMA) as well as a grant to NTR from the
Trottier Institute for Science and Public Policy at McGill University. We thank Dr. Kelly A.
Nugent for collecting and processing the data from the BDB site.




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
