# Peer review of "Simulating soil atmosphere exchanges and CO₂ fluxes for a restored peatland"

_EGUsphere, 2024_

## Referee Comment (RC2)

The study evaluates the potential of restored peatlands to function as CO2 sinks under changing climate conditions with adapting the CoupModel considering the ecohydrological feedback – to simulate land-atmosphere exchanges, including CO2 fluxes at a bog in eastern Canada. After being extracted for eight years, the site was restored by the MLTT (moss layer transfer technique). The model outputs were validated against three years of eddy covariance (EC) measurements of NEE, surface energy fluxes and associated environmental parameters (time since restoration: 14-16 years). The authors also assessed model sensitivity to climate variabilities and moss thickness and compared the emissions factors (EFs) resulted from this study with IPCC Tier-1 EFs. Results showed that the restored peatland has a similar mean CO2 uptake rate to pristine sites, though with higher interannual variability. In an attempt of considering future climate change scenarios, they evaluate the potential CO2 uptake function using the 28-year simulation as a longterm reference run and the possibility of a temporarily shift of the site back to a CO2 source, during dry conditions. The model predicts a moderate reduction in CO2 uptake with future climate change but maintains a CO2 sink function if ecological and hydrological conditions are stable. They show that the CoupModel accurately represents CO2 fluxes, hydrology, and heat exchanges in restored peatlands, based on the findings in this study. They also argue that earlier studies - based on modeling results and IPPC default EFs - might significantly underestimate the climate colling effects for Canadian bogs being restored by MLTT. The authors confirm that restored peatlands exhibit less resilience to climate variability compared to pristine peatlands. The model estimated an emission factor of -1.01 ± 0.64 t C ha$^{-1}$ yr$^{-1}$ for Canadian bogs restored via the MLTT, almost five times the IPCC Tier-1 default emission factor. To summarize, the authors demonstrate that CoupModel simulates the coupled Carbon-hydrological processes in a fine temporal resolution and is now capable of simulating all stages from pristine peatlands to drained for extraction and restoration.

Overall, the manuscript represents an important contribution to scientific progress within the scope of this journal, so the methods - while it has been previously used for different peatland categories - have a broad application and presents new concepts and ideas. The quality of the scientific approach, here in the study, is excellent and the applied methods are valid. The results are discussed in an appropriate and balanced way and related work, including appropriate references, are considered. The results and conclusions are presented in a clear, concise, and well-structured way. The visualization and the quality of figures and tables are good except for a couple of issues raised below.

Please see the comments below. I would like to receive the revised manuscript prior to acceptance.

General comments: While the manuscript provides valuable insights into peatland carbon dynamics by using site-specific EC measurements to validate the model, drawing broad conclusions across all restored peatland types based on data from a single site may not be an ideal scenario. Peatlands exhibit considerable spatial and temporal heterogeneity, with carbon fluxes influenced by factors like hydrology, vegetation composition, peat depth, climate, previous land use and land management history. Validating the model with data from additional sites representing a range of peatland categories and environmental conditions would enhance confidence in the model's applicability to different peatland types. The authors

might consider discussing the limitations of single-site data and suggesting avenues for further validation across a broader dataset to be representative of different categories and to strengthen the robustness of their conclusions for various peatland ecosystems.

Title: To enhance clarity and avoid confusion between soil-atmosphere exchanges and ecosystem-atmosphere exchanges, I would suggest to revise the title. The title must cover the broader scale of exchanges that include the entire ecosystem or land surface and not only soil components. The key processes discussed in this manuscript include NEE, GPP, ER based on various meteorological data and vegetation types or i.e., the thickness of the of the newly grown mosses and not necessarily on soil processes. Therefore, to avoid confusion on presenting the dominant role of i.e., heterotrophic respiration mainly through microbial activities, a broader ecosystem level exchange encompassing the entire land surface (soil, vegetation, water bodies) is recommended for the title. Furthermore, "CO2 fluxes" is already a component of land-atmosphere exchange, I am not sure what the purpose of bringing "... **AND** CO2 fluxes" for a restored peatland is?

Another general comment is referring to "C fluxes" term throughout the manuscript. The authors need to clarify whether the C source-sink transition here means only C from CO2 or CH4 fluxes are also considered. If CO2 is only the focus of the study, then all the C sink strength or uptake function has to be reviewed and changed to C-CO2 or CO2 (for instance: line 30 & 32). Or in line 55-58: it is stated first about the CO2 status of the systems and then switch to C sinks and uptake rate. This has to be determined and specified throughout the manuscript.

L59: to a great extent and not fully.

L135-138: please indicate the peat depth as it can be an indicator of disruption component of the hydrological and temperature balance needed for moss survival and peat formation in such study sites.

L149-50: While it has been referred to the original reference, it is important to provide more details in this section regarding the adapted model parameters, forcing components, etc. and refer to the supplementary table S1.

L165-178: Please consider a review of the vegetation composition listed here and the three PFT groups and confirm whether the vegetation coverage percentage associated with the predominant wind direction matches with Nugent et al. 2018 survey or not.

L185: I would suggest "aerodynamic conductance for both heat and momentum transfer".

L276-278: This statement needs to be elaborated better and mentioned earlier in the text as one of the MLTT issues to be considered.

L280: Maybe better with "Impact of interannual climate variability on $CO_2$ uptake in restored peatlands"?

L287: Shown in any figures or tables?

L295: Unfortunately, this figure is not informative as the years are not separated.

L295-299: Does this mean the $CO_2$ sink strength was mainly correlated with precipitation and Tair shows a non-significant correlation with $CO_2$ uptake? Any other correlation analysis done?

L315: Maybe better with "Impact of future climate change on $CO_2$ uptake in restored peatlands"?

L324-327: This should be elaborated in the discussion and compared better with an earlier correlation analysis where the Tair shows a non-significant correlation with $CO_2$ uptake.

L327: Decrease of what? Please be clear on $CO_2$ uptake decrease or something else?

L336-338: This is a strong statement which is not supported by previous studies. This study in only based on a model validated against 3 years of in-situ measurements. The statement cannot apply to all restored peatlands (even though restored and rewetted must be interpreted differently) with different vegetation composition, previous land use, water table dynamics or altered hydrology. Therefore, I would recommend to rephrase this sentence and make it specific for this type of peatlands i.e., bogs and not applicable to all peatland types. Since no site-specific response is considered here, different peatland types (e.g., bogs vs. fens) and stages of restoration might respond differently to these changes. The statement implies a uniform response, which may not be accurate. Also, you are discussing "extreme" scenarios earlier in this paragraph; do you mean future climate change? Climate extreme events could potentially lead into the vegetation die-off if the extremes, such as drought, are persistent (long-enough) and severe. The statement also implies that restored peatlands will remain unaffected by climate change, which is unlikely. A better term might be "may retain some capacity for $CO_2$ uptake," reflecting that while uptake might continue, it could be reduced or fluctuate significantly.

L367-368: This might vary across sites and not necessarily reach a stage of fully recovered vegetation after 14-16 years, depending on site-specific characteristics. The analyses here and earlier are based on the

system behavior 14-15 years after restoration. Could you provide some data for the earlier years after restoration (year 1 to year 14) to compare the CO2 trajectories immediately after restoration and one decade later? I was curious whether there were any other measurements than EC data from that specific area (within the footprint of the tower); i.e., I guess chamber measurements were conducted there? Any comparisons done?

L377-387: While this is relevant information, it is difficult to make the link and visualize them. Please elaborate further on this or add data to support this discussion. Also, another general comment: perhaps including a high-resolution map/image of the site, with footprint isolines, could greatly enhance readers' understanding by linking land cover types with the associated fluxes. This visual context would allow readers to better interpret the spatial relationship between the site's land surface characteristics and flux measurements.

L396: t C-CO2 ha$^{-1}$ yr$^{-1}$ should be mentioned.

L404-405: While MLTT seems to be beneficial for the recovery of peatland C uptake, we would need to always consider that mosses need consistently high water levels to thrive after MLTT restoration. Therefore, an altered natural hydrological condition of such systems requires to maintain a stable WTD and that can be challenging, especially in areas where drainage has lowered groundwater levels and led to a less resilient system. Also, if water level drops even temporarily, especially during dry periods or in warmer climates, this can lead to poor establishment and even die-off of mosses, reducing restoration success. So, impact of future climate extremes has to be mentioned here. Furthermore, the influence of nutrient and altered pH levels - that can encourage invasive species outcompeting mosses - were not considered. I noticed that you've touched this issue earlier indicating "Since these are oligotrophic ecosystems, the influence of nutrients on C was not considered in this study".

L407-408: Is this the main driver?

L455: Is there any indication of the trends in NPP mosses throughout the manuscript text?

L465: The figures here are not informative as the years and interannual variability are not well-visualized.

L477-480: This concerns the code only and not data availability.

L491 & 493: NTR, the same as NR, I assume?

Supplementary:

L29: A figure showing the climatic water balance (CWB) – the difference between precipitation and evapotranspiration (ET) or potential ET (ET0) – and water table depth would be useful for readers' interpretation.

---

## Community Comment (CC2)

We appreciate the comments made by Aldis Butlers for our manuscript.

Dr. Butler commented on the model evaluation. In our paper we evaluated CoupModel with three years of eddy covariance fluxes and supported environmental data for a restored peatland. We acknowledge that other chamber measurements exist elsewhere, and these might be useful for further model evaluation in the future. Note detailed model evaluation results were presented in Fig. 1 and 2. Model sensitivity analysis were made with regards to the thickness of the newly grown acrotelm (Fig.3) and climate (Fig. 6,7 and 8). The parameter sensitivities of CoupModel for pristine bog, and bog undergoing extraction were addressed in earlier studies (supplement section F of He et al. 2023, Hydrology and Earth System Sciences and e.g. Table 3 of He et al. 2023, Ecosystems). The simulated NEE is compared with measured NEE (Fig. 1i, 2), also see Line 252-260.

The study site is oligotrophic, raised bog typical of continental North America.  The restoration approach used for the study site is well documented (e.g. Quinty and Rochefort, 2003; Gonzalez and Rochefort, 2014) and several earlier studies investigated the sites nutrient status (e.g. Andersen et al. 2010). The Canadian practices require the extracting companies to leave a residual peat layer to facilitate restoration using the moss-layer-transfer-technique (MLTT). In BDB, two meters with 80 cm *Sphagnum* peat on top was left over in 1999 before the restoration. Hence, there is no groundwater altering nutrient conditions in our study site. We will add more site description to make this clearer in our revision.

**References**

González, E. and L. Rochefort (2014). "Drivers of success in 53 cutover bogs restored by a moss layer transfer technique." Ecological Engineering 68: 279-290.

He, H., Moore, T., Humphreys, E. R., Lafleur, P. M., and Roulet, N. T.: Water level variation at a beaver pond significantly impacts net $CO_2$ uptake of a continental bog, Hydrol. Earth Syst. Sci., 27, 213–227, https://doi.org/10.5194/hess-27-213-2023, 2023.

He, H., Clark, L., Lai, O.Y. *et al*. Simulating Soil Atmosphere Exchanges and $CO_2$ Fluxes for an Ongoing Peat Extraction Site. *Ecosystems* **26**, 1335–1348 (2023). https://doi.org/10.1007/s10021-023-00836-2

Andersen, R., Rochefort, L. & Poulin, M. Peat, Water and Plant Tissue Chemistry Monitoring: A Seven-Year Case-Study in a Restored Peatland. *Wetlands* **30**, 159–170 (2010). https://doi.org/10.1007/s13157-009-0015-0

Quinty, F. and Rochefort, L.: Peatland Restoration Guide: Second Edition, Canadian Sphagnum Peat Moss Association and New Brunswick Department of Natural Resources and Energy, 2003.

---

## Author Comment (AC1)

We would like to thank both reviewers (Dr. Jentzsch and anonymous reviewer 2) for their constructive criticism and comments that helped us improve our paper. In following we address the general comments raised and for minor comments we will take those into consideration during our final revision of the paper.

First, both reviewers commented the first two conclusions drawn from this study and its broad applicability. We agree that our study is based on a specific peatland system: continental bogs which covers majority of Canadian extensive peatlands (e.g. Webster et al. 2018, https://doi.org/10.1186/s13021-018-0105-5), thus drawing general conclusions to all peatland types should be avoided. In our revision we will rephase the wording to avoid confusion. However, note we have specifically discussed the future need to work on other peatland types, e.g. Line 391-393 and the need to study other restoration stages, e.g. Line 367-370. We also agree on the suggestions from both reviewers on revising the title to be clearer and add more details about the model descriptions.

**On comments from Dr. Jentzsch (reviewer 1)**

Dr. Jentzsch further suggested to elaborate structural changes in a restored peatland with time in the introduction section and move the explanations of model data discrepancies into discussion. We believe these are good ideas and will revise accordingly.

Specifically, they commented on the rationale of the selection of new acrotelm thickness 20, 40, 100 cm. These were designed to test the model sensitivity to different ecohydrology settings, representing a varying stage of post restoration. The acrotelm thickness was 30 cm in 2013 when the flux measurements started, thus the 20, 40 cm were selected by +-10 cm. Field data at BDB from earlier publications (e.g. McCarter and Price, 2015, https://doi.org/10.1002/eco.1498) suggests ~2009 (c.a. 10 years after restoration) the new acrotelm is ~15-20 cm thick, we would expect that with a decadal or two 40 cm thickness would be reached. Thus, the testing range 20-40 cm roughly represents ~10 to ~30 years of post restoration. Another reason for selecting 40 cm is because the pristine peatland, Mer Bleue, another continental bog in Canada with an extensive research, has an acrotelm of ~40 cm (defined by average of long-term water table depth, see He et al 2023 HESS, https://doi.org/10.5194/hess-27-213-2023). The selection of 100 cm acrotelm thickness is hypothetical but rather used to demonstrate the importance of mestelm collapse layer in supporting the moss growth, and to validate/evaluate the ability of the model to reproduce the measured empirical threshold (-100 cm water tension) for moss growth. We have discussed this e.g. Line 381-384. This further shows the robustness of the model to reproduce the ecohydrological controls of the restored site. We are currently doing further research to attempt to simulate the creation of a mesotelm and new catotelm layer on top of the residual peat. This research will evaluate how the compaction/collapse of the

mesotelm layer, by increasing bulk density, reducing porosity and hydrological conductivity of the mesotelm layer will influence the water table depth and water availability for the mosses and other vegetations. However, there are very few, if any restored bogs that are old enough to have developed a thicker restore peat layer than 30-40 cm. We are looking at some older block cut peatlands as an analog for older restored sites.

Dr. Jentzsch also suggested us to compare the simulated long term BDB CO2 flux data with the simulated MB data rather than measured data. While we are unsure if we fully understand the rationale behind this suggestion. The comparison (simulated BDB vs Measured MB) was made to show first the flux magnitude (mean), second the annual variations (S.D.). Our earlier model evaluation at MB showed that the simulated NEE was -67± 51 g C $m^{-2}$ $yr^{-1}$ for 2012-2016, -90 ± 35 g C $m^{-2}$ $yr^{-1}$ for 2004-2012 while the corresponding measured NEE was -102 ± 40 and -115 ± 33 g C $m^{-2}$ $yr^{-1}$, respectively (Fig. 5b in He et al. 2023, https://doi.org/10.5194/hess-27-213-2023). This confirms that CoupModel can reproduce the measured S.D. as the observed MB data. Thus, no difference will make as the current version when we compare the simulated BDB vs simulated MB. Nevertheless, we will add the comparison in our revision.

We agree that the remnant infilled ditches can influence the C uptake functions as Dr. Jentzsch pointed out. Note Nugent et al. 2018 https://doi.org/10.1111/gcb.14449 discussed the influence of remnant ditches on C fluxes at BDB. Their analysis suggested the effect of ditches at the ecosystem level was small as ditches represent a minor fraction of BDB, but a higher $CH_4$ flux was measured when *Typha latifolia*-invaded drainage ditches were in the tower footprint. No clear influence on $CO_2$ was found. We will rephase the C uptake function here to $CO_2$ uptake function to avoid confusions.

Finally, we agree that an additional conclusion from our climate change simulations should be added.

**On comments from reviewer 2**

First Reviewer 2 suggested us carefully checking the terminology of C used in the paper, we fully agree and here only $CO_2$-C is addressed, we will revise that to avoid confusions.

Reviewer 2 also suggested to add details of the study site e.g. peat depth (mean peat thickness is ~2.2 m, and a maximum of 3.75 m), and vegetation composition. We will revise this in our revision.

Note details of flux footprint and vegetation survey distribution and results were given in Nugent et al. 2018 https://doi.org/10.1111/gcb.14449. Their footprint analysis revealed the

restoration area was classified as 96% restored field and 4% infilled ditches. The restored section was surrounded by forested peatland which limits fetch to 200 m toward the west, 150 m toward the north and south, and 100 m toward the east (abuts an unrestored section). The dominant wind direction was west and north, but 30% comes from the south from August to December. Seldom is the wind direction from the east (ECCC, 2023 station ID: Riviere Du Loup). Table 1 from their paper (in Nugent et al. 2018 https://doi.org/10.1111/gcb.14449) shows the vegetation survey results and its distributions. Note the vegetation distribution in the field is quite homogenous across the major survey direction. There is difference for the remnant ditches and the field. However, since only 4% surface is covered by the ditches, and an average approach is used in CoupModel by using results of "all directions" (term used in their Table 1) to initialize the vegetation cover. Thus, we believe the influence of vegetation composition with tower footprint on our simulation results are minor. Unfortunately, a detailed footprint map is not available, but we will add a few sentences to motivate our vegetation initialization in the revision.

Table 1 Percent vegetation cover and ditch cover of BDB for three 30° direction bins for the area of the mean growing season 80% probability tower flux footprint, taken from Nugent et al. 2018 https://doi.org/10.1111/gcb.14449

| Physiographic feature | Functional type | 30–60° (NE) | 200–230° (SW) | 290–320° (NW) | All directions |
|---|---|---|---|---|---|
| Field | Vascular | 66 | 76 | 72 | 75 |
| | Ericaceous shrubs | 36 | 42 | 42 | 39 |
| | Sedges | 27 | 35 | 26 | 33 |
| | *Typha latifolia* | 0 | 1 | 10 | 0 |
| | Non-vascular | 55 | 74 | 51 | 69 |
| | *Sphagnum* | 50 | 67 | 33 | 61 |
| Ditch | Vascular | 85 | 83 | 73 | 85 |
| | Ericaceous shrubs | 63 | 53 | 41 | 51 |
| | Sedges | 30 | 23 | 25 | 29 |
| | *Typha latifolia* | 0 | 6 | 19 | 6 |
| | Non-vascular | 41 | 39 | 57 | 44 |
| | *Sphagnum* | 14 | 7 | 2 | 8 |
| $FC_{ditch}$ | | 7 | 2 | 4 | 4 |

Reviewer 2 commented on the correlation analysis (see L 295-299) between the simulated annual $CO_2$ fluxes and the annual mean climate variables, i.e. precipitation and air temperature. The results show both non-significant correlations, but air temperature showed a slightly higher correlation coefficient. We will rephase this to make it clearer.

Reviewer 2 also suggested to provide some data for earlier years after restoration. We had provided that in the introduction section, see Line 56. These measurements were made right after the restoration and showed a source to the atmosphere 200-500 g C m$^{-2}$ yr$^{-1}$ (Petrone et al 2001, https://doi.org/10.1002/hyp.475)

We agree on the comments made on future climate extremes, nutrients and pH effects should be included in the future studies on L404-405, thus will add it into our revision.

Comments on Line 455, note we have described the trend of NPP moss in L. 276-278.

Finally, Reviewer 2 commented on the figures S3, S4 in the supplement, suggesting showing the climatic water balance – difference between precipitation and ET. We agree and will do that accordingly.

---

## Editor Decision (ED1)

**Simulating  ecosystem carbon dioxide  fluxes and their associated influencing factors for a restored peatland**

Hongxing He[1]; Ian B. Strachan[2]; and Nigel T Roulet[1]

Hongxing He, https://orcid.org/0000-0003-4953-7450

Ian Strachan, https://orcid.org/0000-0001-6457-5530

Nigel T Roulet, https://orcid.org/0000-0001-9571-1929

[1] Department of Geography, McGill University, Montréal, Quebec, Canada

[2] Department of Geography and Planning, Queen's University, Kingston, Ontario, Canada

Correspondence, HH hongxing.he@mcgill.ca; hongxing-he@hotmail.com

**Abstract**

Restoration of drained and extracted peatlands can potentially return them to carbon dioxide ($CO_2$) sinks, thus acting as significant climate change mitigation. However, whether the restored sites will remain  sinks or switch to sources with a changing climate is unknown. Therefore, we adapted the CoupModel to simulate ecosystem $CO_2$ fluxes  and the associated influencing factors  of a restored bog. The study site was a peatland in eastern Canada that was extracted for eight years and left for 20 years before restoration. The model outputs were first evaluated against three years (representing 14-16 years post restoration) of eddy covariance measurements of net ecosystem exchange (NEE), surface energy fluxes, soil temperature profiles, and water table depth data. A sensitivity analysis was conducted to evaluate the response of the simulated $CO_2$ fluxes to the thickness of the newly grown mosses. The validated model was then used to assess the sensitivity  to changes in climate forcing. CoupModel reproduced the measured surface energy fluxes and showed high agreement with the observed soil temperature, water table depth, and NEE data. The simulated NEE varied slightly when changing the thickness of newly grown mosses and acrotelm from 0.2 to 0.4 m but showed significantly less uptake for a 1 m thickness. The simulated NEE was $-95 \pm 19$ g C m$^{-2}$ yr$^{-1}$ over the three evaluation years, and $-101 \pm 64$ g C m$^{-2}$ yr$^{-1}$, ranging from $-219$ to $+54$ g C m$^{-2}$ yr$^{-1}$ with an extended 28-year climate data. After 14 years of restoration, the peatland has a mean $CO_2$ uptake rate similar to pristine sites, but with a much larger interannual variability, and  in dry years, the restored peatland can switch back to a temporary $CO_2$ source. The model predicts a moderate reduction of $CO_2$ uptake, but still a reasonable sink under future climate change conditions if the peatland is ecologically and hydrologically restored. The ability of CoupModel to simulate the $CO_2$ dynamics and its thermal-hydro drivers for restored peatlands has important implications for emission accounting and climate-smart management of drained peatlands.

**Keywords**: Restored peatland; climate variability; net ecosystem exchange; water table depth; emission factor; simulation

**1 Introduction**

Degradation of peatlands through land use change and drainage is currently estimated to emit ~ 4% of global annual anthropogenic carbon dioxide (United Nations Environment Programme, 2022). Therefore, restoring drained peatlands so that they return to carbon (C) sinks has been identified as an emerging priority for climate change mitigation (Leifeld and Menichetti, 2018). When ecologically restored successfully, peatlands can generally return to their carbon (C) uptake function after a decade or two following the recolonization of peatland vegetation and a decrease in water table depth (Nugent et al., 2018; González and Rochefort, 2014; Richardson et al., 2023; Tuittila et al., 1999; Wilson et al., 2016; Beyer and Höper, 2015). However, the C uptake function of restored peatlands is sensitive to climate conditions, particularly in drier years (Wilson et al., 2016). Therefore, changing climate can potentially weaken the sink strength or even switch the restored peatlands into C sources.

In North America, about a quarter of drained peatlands that were earlier used for horticultural peat extraction have been restored by the Moss Layer Transfer Technique (MLTT) (Chimner et al., 2017; Quinty and Rochefort, 2003). Ecosystem-scale flux measurements indicate peatlands remain a $CO_2$ source (~200 to 500 g C m$^{-2}$ yr$^{-1}$) the first few years of restoration (Petrone et al., 2001; Petrone et al., 2003), but after a decade or two, peat vegetation recovers, and the restored bogs return to $CO_2$ sinks with uptake rates similar to pristine sites (Nugent et al., 2018). While the C accumulation function can generally be fully restored within a decade or two, full restoration of the peat soil structure and ecohydrology takes a much longer time (Loisel and Gallego-Sala, 2022) with centuries to millennia required for the restored peatland to accumulate the C that was extracted. Restoration creates a novel ecosystem in transition to a rewetted steady state and the altered ecohydrology decreases peatland ecological resilience (Kreyling et al., 2021).

The ecological function of peatlands is strongly linked to ecohydrology (Waddington et al., 2014).  Clymo (1992) outlined four functional layers of pristine peatlands (i.e. green- peat litter- collapse- peat proper, Fig. 1 in his paper) and how the peat structure interacts with ecohydrology, thus regulating the growth and function of peatlands. Briefly, the bulk density of the green and peat litter layer is low, typically below 0.05 g cm$^{-3}$. The increasing load of new growth above and the mass proportion of water, as well as the decomposition of plant material causes the moss structure to collapse, typically increasing the bulk density gradually along the peat profile to ~0.1 g cm$^{-3}$. The result is a reduction in the space between dead leaves and stems and the soil pore sizes, increasing the capillary force for vertical water movement, thus sustaining the water supply for sphagnum mosses and the growth of the peatlands.  For extracted peatlands, the MLTT gives a jump start for mosses colonization at the residual catotelmic peat surface, with time, a new layer of acrotelm is formed and thicken. However, these newly regenerated moss with low bulk density forms large pores directly above the dense residual peat remaining after extraction (catotelmic peat) and does not have the negative interstitial pressures required to draw pore water, causing a capillary barrier effect (Gauthier et al., 2022; Gauthier et al., 2018). The capillary barrier decreases the ability of the new moss to draw water from the deeper compacted catotelmic peat, thus causing an overall lower surface moisture content for restored sites compared to natural peatlands (McCarter and Price, 2015). As a result, the new moss layer may become stressed quickly and even die off during dry periods. Synthesis studies have shown that vegetation colonization is much slower after restoration over warm and drier years (González and Rochefort, 2014), and data from a restored Irish extracted bog show a less resilient C uptake function over the drier years (Wilson et al., 2016).

Under the United Nations Framework Convention on Climate Change (UNFCCC), countries with peatlands managed for extraction are required to report greenhouse gas emissions annually (IPCC, 2014).  Currently National Inventory Report (NIR) of Canada report emissions from restored peatland separately and an emission factor (EF) of +2.07 ton [tonnes or t] $CO_2$ -C

ha$^{-1}$ yr$^{-1}$ (positive meaning source) generated from data of three sites (all restored less than 10

years) is used (ECCC, 2021). However, the emission changes [CO2 emissions change] with time as the peatland development [develops] and gradually switch to an uptake [switching to CO2 uptake]

(Nugent et al., 2018).

[revised manuscript text omitted]
 (~10 years after restoration), 40 cm (~30 years after restoration), and 100 cm (hypothetical, to test the empirical threshold of -100 mb). For the latter two model simulations, peat properties of the 20-30 cm layer in the reference run (Table 2) were assumed for the future extra 10 and 70 cm acrotelm, respectively. The vegetation was assumed to be the same as the reference run and the peat compaction due to the growth of mosses and decomposition was not considered (i.e., no mesotelm collapse layer).

Our sensitivity analysis showed that the simulated NEE uptake increased slightly when changing the new acrotelm thickness from 20 to 40 cm but reduced (meaning less uptake)

significantly for the model run with an acrotelm of 100 cm (Fig. 3). The small changes of simulated NEE can be explained by both increase of GPP and ecosystem respiration (ER) with increasing new acrotelm thickness (20-40 cm). The NPP (net primary production) of mosses show a slight decrease trend with increasing acrotelm thickness (Fig. 3). The reduction of $CO_2$

uptake in the 100 cm acrotelm thickness model run is because the model simulated that the surface mosses start to die off because they cannot take up water from the deep peat (Fig. 3).

Type text here

**3.3 Impact of interannual climate variability on $CO_2$ uptake of restored peatlands**

[revised manuscript text omitted]

10 years thus explaining the current EF, +2.07  tonnes $CO_2$ -C ha$^{-1}$ yr$^{-1}$ used in Canada NIR. We argue this EF  do not reflect the temporal dynamics of greenhouse gases for restored peatlands, particularly for those sites that have fully vegetation recovery (Kalhori et al., 2024).

Our 28-year extended model simulations by considering the interannual climate variation suggest an EF of -1.01 ± 0.64 t $CO_2$-C ha$^{-1}$ yr$^{-1}$ for a bog 14-16 years post restoration by the MLTT. This should be included in the next revision of EF within NIR of Canada.

Moreover, our modelled EF is ~five times larger (meaning more uptake) than the default

IPCC Tier 1 EF for temperate nutrient-poor rewetted organic soils (-0.23 with CI -0.64 to +0.18

t C ha$^{-1}$ yr$^{-1}$ (IPCC, 2014). The data used to generate the IPCC EF includes more degraded sites in Europe and different rewetting methods. The Canadian practice of leaving a residual peat layer at the end of extraction and using MLTT for restoration seems to be beneficial for the recovery of peatland C uptake. The default IPCC EF has earlier been used to evaluate the overall climate impacts for peatland restoration using a modeling approach (Gunther et al.,

2020). Our results thus suggest those studies might significantly underestimate the climate cooling effects for Canadian bog sites that have been restored using MLTT.

Our climate change simulations show the  regulating affect of air temperature on the

$CO_2$ uptake of restored peatlands, where future global warming is predicted to moderately weaken the sink strength (Fig. S3). However, it should be noted that future changes in seasonal patterns and extremes were not accounted for in our climate change scenarios. Helbig et al.

(2022) analyzed flux measurements from northern peatlands and showed earlier summer warming increases NEE uptake while late summer warming decreases it. The seasonal patterns and particularly extremes of climate can be additional factors  Controlling the $CO_2$ fluxes. For mosses and peatland vegetations to develop, a stable water table is required. However, this can be challenging under the altered ecohydrological and climate condition, especially in areas where drainage has lowered groundwater levels causing less resilience of the ecosystem. Our results show even water level drops temporarily, especially during dry periods or in warmer climate this can lead to poor establishment and even die off mosses, reducing the restoration success consequently ecosystem uptake of $CO_2$. Helbig et al. (2022) analyzed flux measurements from northern peatlands and showed earlier summer warming increases NEE uptake while late summer warming decreases it. In addition, Tthere is also the possibility of fire that would structurally alter the peatlands. Our simulations do not include fire, which is much less common in eastern Canadian peatlands than in the west (Zoltai et al., 1998; Lavoie and Pellerin, 2007). Thus, our climate change simulations probably overall represent a conservative prediction which might in turn explain the moderate reduction of sink strength. As our extended simulations show, it is possible that over extreme years the site can switch to a small $CO_2$ source and that potentially the number of source years could increase in the future.

**5 Conclusion**

This study applied the CoupModel to a peatland site restored 14-16 years previously. We conclude:

- CoupModel can describe the measured sub-daily $CO_2$ fluxes, hydrology, and heat for the restored peatland bog system.

- Restored peatlands bogs have less resilience to climate variability than pristine bog systems peatlands.

- CoupModel simulation results in an emission factor of -1.01 ± 0.64 t C ha$^{-1}$ yr$^{-1}$ for Canadian bogs that have been restored for 14 to 16 years by the moss layer transfer technique, ~ five times larger than the IPCC default emission factor and much smaller than current emission factor used in the Canadian NIR.

• Moderate reduction of $CO_2$ uptake is predicted for restored bogs with fully vegetation cover under future climate change conditions.

[revised manuscript text omitted]

---

## Author Response (AR2)

CLASSIFIED - NON CLASSIFIÉ

**Further Point to Point Reply to referee comments on egusphere-2024-2679**

*We would like to thank anonymous reviewer 2 and editor for their additional comments that helped us further improve our paper.*

*We have now incorporated all the suggested text editions by reviewer 2 and revised the figure 5 into a colour figure, make it larger to increase the readability, and add the legends to show each individual year.*

*We also revised the Line 445-447 in discussion to make it more clear.*